# Sleep Quality in Older Women: Effects of a Vibration Training Program

**María Victoria Palop-Montoro [1], Emilio Lozano-Aguilera [2], Milagros Arteaga-Checa [3]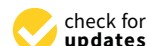, Víctor Serrano-Huete [3], Juan Antonio Párraga-Montilla [3] and David Manzano-Sánchez [4],***

1   Department of Physiotherapy, Faculty of Health Sciences, Universidad Católica San Antonio de Murcia, 30107 Murcia, Spain; mvpalop@ucam.edu
2   Department of Statistics and Operations Research, Faculty of Social and Legal Sciences, Universidad de Jaén, 23071 Jaén, Spain; elozano@ujaen.es
3   Department of Musical, Plastic and Body Expression, Faculty of Humanities and Educational Sciences, Universidad de Jaén, 23071 Jaén, Spain; marteaga@ujaen.es (M.A.-C.); victor.serrano@ui1.es (V.S.-H.); jparraga@ujaen.es (J.A.P.-M.)
4   Department of Physical Activity and Sport, Faculty of Physical Activity and Sport Sciences, Universidad de Murcia, 30720 Murcia, Spain
*   Correspondence: david.manzano@um.es; Tel.: +86-888-8671

**Abstract:** Background: Sleep is an important phenomenon to restore the body, both physically and emotionally, providing a state of balance in the person. It has been proven that adequate sleep at night is one of the main needs of older people in order to maintain an active and healthy life; among other factors, regular physical exercise can improve the quality of sleep. The aim of this research is to evaluate the effects of a physical exercise program supplemented with vibration training on sleep quality and the use of sleep drugs in women over 65 years of age. Methods: Fifty-two independent, physically active adult women were randomised into two groups: a physical exercise program (n = 26, control group) and the same physical exercise program supplemented with vibration training (n = 26, experimental group). The control group performed two weekly sessions of exercise, and the experimental group added another two sessions of vibration training to these two sessions for 12 weeks. Sleep quality was assessed using the Pittsburgh index. Statistical significance was established as $p < 0.05$. Results: After the intervention, there were significant changes to the quality of sleep ($p = 0.001$) and hours of sleep ($p = 0.002$) in the experimental group. The consumption of drugs decreased in this group, although not significantly; however, it did have a moderate effect size ($p = 0.058$; d = 0.36). The control group, on the other hand, reported significantly worsened sleep quality ($p = 0.001$) and increased drug use ($p = 0.008$). Conclusion: Three months of vibration training, as a complement to a conventional physical exercise program, improves sleep quality and reduces the consumption of sleeping pills in women over 65 years of age.

**Keywords:** whole-body vibration; physical exercise; quality of sleep; health; quality of life; ageing; drugs

---

## 1. Introduction

Sleep is a state of physiological rest characterised by its periodicity and reversibility. It is considered an important phenomenon to restore the body both physically and emotionally [1], providing balance in the person. The sleep pattern of each individual differs enormously from that of others, either by duration or structure, throughout the day; in addition, other factors can decisively influence its quality, such as level of economic status, place of residence, nutrition, and sedentary lifestyle [2]. Research

shows that a healthy diet rich in fruits, vegetables, and olive oil improves the quality of sleep in older adults [3–5]. Other factors, such as stress or psychosocial factors, can also have a great influence on sleep [6–8]. With ageing, this pattern changes in a special way, not affecting the total sleep time, but by being interrupted and more distributed throughout the day [1].

It has been proven that adequate sleep at night is one of the main requirements of the elderly to maintain an active and healthy lifestyle [9]. Royuela et al. [10] found that there is a 54% prevalence of poor sleep quality in those over 65 years of age, indicating that more than a third of older people have sleep problems and up to a quarter have serious difficulties. According to the American Psychiatric Association [11], chronic sleep disturbance is a risk factor for the subsequent onset of anxiety and mood disorders. Additionally, a recent study by Serrano-Checa et al. [12] associated anxiety, coupled with poor sleep quality, with the risk of falls and joint problems in women over 60 years of age.

Few people seek help with their sleeping difficulties and those who do often seek help after years of experiencing discomfort from this situation [13]. In this sense, there has been a complex debate since the same authors indicate that there are people without real impairment who complain of sleep problems and another group of people with true signs and symptoms who do not report their rest-related disorders.

Physical exercise, as a nonpharmacological intervention, can be an effective strategy with which to improve the quality of sleep. Specifically, it has been shown that replacing a sedentary lifestyle with light-intensity exercise could have long-term benefits on the quality of sleep in older people [14,15]. The benefits of different forms of physical exercise on the quality of sleep have been demonstrated [16–18]. Likewise, the regular practice of exercise seems to be a protective factor of cognitive function, with the quality of sleep being a decisive factor in reducing the symptoms of depression [19]. Similarly, physical exercise and quality of sleep have consistently been associated with quality of life in clinical and nonclinical populations. However, the mechanisms underlying this relationship are not well understood. What is known is that sleep quality directly influences the quality of life through physical, mental and social wellbeing and that physical exercise also indirectly predicts quality of life and quality of sleep [20].

An option that, for some years, has been introduced with the aim of improving the physical performance of both athletes and sensitive populations is vibration training, also called whole body vibration (WBV). The mechanical stimulus that this vibration transmits, through oscillatory movements, increases the gravitational load to which the neuromuscular system is subjected. Although parameters such as duration of application, frequency, and wave amplitude are important, they must be taken into account according to the functional capacity of each person [21]. Melloni et al. [22] indicated that the vibration increases the feeling of relaxation after its performance. Various studies have indicated that WBV can produce positive effects in different age groups. In older people, the effects of vibration training on different body systems have been proven in numerous studies in bone mineral density [23], body balance, muscle performance, preventing falls [24], and quality of life [25]. This type of training can increase the recruitment of fast-twitch muscle fibres and delay ageing-induced muscular dystrophy, thus improving muscle strength and power [26]. Therefore, it may be beneficial to reduce sarcopenia and increase the physical capacity of this group [27,28], although there is a discrepancy regarding the improvement of sleep quality [28].

Scientific evidence has shown that WBV produces positive effects on various bodily functions [29] and can be considered an effective therapy for the treatment of chronic diseases in the elderly [30]. Vibration causes a neurogenic effect of vasodilation, increasing blood flow to the muscles and reducing their acidification [31,32]. It also influences the production of hormones [33] and reduces the consumption of drugs in some age-related diseases [34]. Research suggests that WBV may positively affect muscle performance [35,36]. However, other studies have reported equivocal results [37,38]. The diversity of vibratory training protocols employed makes the comparison of interventions extremely difficult.

On the other hand, the consumption of drugs to induce sleep has also been the object of study, carrying out different interventions to reduce its consumption, although most studies focus on cognitive or cognitive–behavioural therapies [39], self-help therapies [40], or therapies related to physical exercise [41–44]. Instead, few studies have analyzed the influence of physical exercise combined with vibratory training on sleep quality; the results are not conclusive in the group of older people. Therefore, the objective of this study is to evaluate the effects of a regular physical exercise program, complementing it with vibration training, on sleep quality, as well as to analyze whether this type of exercise reduces the need for drug use to induce sleep in women over 65 years of age.

Thus, a physical exercise program complemented with vibratory training constitutes our intervention proposal. Although the two strategies in isolation seek clinical success, they are fields of action for health professionals in the sports and therapeutic field. In the hypothesis of this study, we suspect that the clinical impact on the ageing processes of a proposal of physical exercise for 12 weeks, twice a week, is better supplemented with vibration training than without it on the quality of sleep and can reduce the consumption of sleeping drugs in women over 65 years of age.

## 2. Materials and Methods

An experimental, longitudinal study was carried out for 12 weeks, with pre- and post-test measurements, intervening through a physical exercise program together with WBV to assess its impact on the quality of sleep in older women.

### 2.1. Participants

The sample initially consisted of 56 healthy adult women, the users of two day-centres in Jaén (Spain), aged between 65 and 80 years. As inclusion criteria, it was established that they were women between 65 and 85 years old, being independent in their activities of daily life, attending the physical exercise program, committed to complying with the protocol established during the research, having accepted the rules of the organization and the methodology, as well as completed all of the sessions. The exclusion criteria were the following: not having medical authorization to participate in the program or suffering from a disease that makes it impossible to carry out any type of physical activity. The participants provided us with a medical report of their personal history and current pathology. Women with acute or decompensated heart and respiratory diseases were not included in this study.

The participants were distributed into two groups: a control group (CG; mean age = 70.04 ± 4.39; mean BMI = 30.23 ± 4.78) and an experimental group (EG; mean age = 70.58 ± 5.70 years; mean BMI = 28.81 ± 3.93). CG performed a physical exercise program, while EG, in addition to the physical exercise program, performed additional vibration training. EG and CG were defined following a criterion of personal preference for the practice or not of vibration training. Both groups were satisfied, with a balance in terms of the number of subjects (having checked the homogeneity of the groups), so the preference was respected.

It was established as a requirement that the participants had to complete all the sessions. We report that four women in the control group did not complete all the sessions and, therefore, were excluded from the final analysis. A total of 52 women (age: 70.31 ± 5.04 years, weight: 69.92 ± 11.15 kg, height 153.57 ± 6.62 cm) completed all the sessions, which made up the final sample (Table 1).

**Table 1.** Sociodemographic characteristics of the sample.

|  |  | Control | | Experimental | | Total | |
|---|---|---|---|---|---|---|---|
|  |  | **n** | **%** | **n** | **%** | **n** | **%** |
| Civil_State | Single | 2 | 3.8% | 3 | 5.8% | 5 | 9.6% |
|  | Married | 14 | 26.9% | 12 | 23.1% | 26 | 50.0% |
|  | Divorcee | 1 | 1.9% | 4 | 7.7% | 5 | 9.6% |
|  | Widow | 9 | 17.3% | 7 | 13.5% | 16 | 30.8% |
|  |  | M | SD | M | SD | M | SD |
| Brothers |  | 3.42 | 2.16 | 1.96 | 1.66 | 2.69 | 2.04 |
| Sons |  | 2.77 | 1.82 | 2.62 | 1.81 | 2.69 | 1.80 |
| Age |  | 70.04 | 4.39 | 70.58 | 5.70 | 70.31 | 5.04 |
| BMI |  | 30.23 | 0.96 | 28.81 | 0.77 | 29.51 | 0.61 |
| Weight |  | 72.40 | 2.11 | 67.5 | 2.23 | 69.92 | 11.15 |
| Height |  | 154.32 | 1.30 | 152.85 | 1.33 | 153.57 | 6.62 |

BMI = body mass index.

*2.2. Procedure*

In the first place, the two participating centres were visited after obtaining prior authorisation from the Consejería of Health and Social Welfare of the Junta de Andalucía and the directors of both centres, holding the first information session to present the characteristics of the intervention program. Attendees were provided with a document with the details of the study and the protocol to be followed, along with an informed consent form.

The second phase involved the preliminary collection (pretest) of the participants' sociodemographic data and the administration of the "Pittsburgh Sleep Quality Index" (PSQI). In turn, an evaluation of the size and body composition of the sample was carried out by three physiotherapists. Finally, the women were randomly assigned to the groups by incorporating them into an Excel sheet and assigning a number to each of the participants.

The third session began with the intervention, with a personal follow-up of each participant included in the research, ending with the data collection protocol (post-test) and the analysis of results. The intervention had a timing of 12 weeks, with 2 weekly sessions of 60 and 80 min for CG and EG respectively. In both cases, a work routine was established. Both groups carried out 22 practical sessions. The EG participants performed vibration training after the physical exercise session (the same day). The physical exercise program was held at 9:00 a.m. and vibration training at 10:00 a.m.

*2.3. Intervention Protocol*

2.3.1. Physical Exercise Program

The exercise program was carried out as a group in both intervention groups, with two weekly sessions of 60 min for each session and structure and intensity progression based on the guidelines of the American College of Sports Medicine [45]. This consists of an initial time dedicated to organisational tasks (five minutes), warm-up (five minutes), aerobic exercise (25 min), strength exercises (10 min), flexibility exercises (10 min) and a cool-down (five minutes). All participants completed a total of 22 sessions. These sessions were directed and supervised by a graduate with a degree in Physical Activity and Sports Sciences, with more than 10 years of experience in the sector. The main researcher attended the physical exercise program to see its development and asked the participants about their level of satisfaction with the exercise performed, possible discomfort, and if they performed any additional physical activity, which we did not recommend for the duration of the study. The person in charge of carrying out the program called the roll at the beginning of the sessions and then gave the researcher the list of participants in each session.

### 2.3.2. Vibration Training

To become familiar with the use of the vibratory platforms, as well as the correct execution of the exercises to be performed on them, the experimental group carried out an initial adaptation program lasting for one week, with two weekly sessions, each lasting 20 min. The participants, divided into two groups of 13, practised every other day with WBV (Monday and Wednesday, one group; Tuesday and Thursday, another group), just as they would in the subsequent training phase. After this adaptation to the methodology and apparatus, 26 women remained in the research.

The WBV protocol consisted of two weekly sessions, held between 10:00 and 11:00 a.m., following the suggestions of Roveda et al. [16], which indicate that physical exercise performed in the morning helps to avoid nocturnal sleep problems to a greater extent and thus improves sleep quality. The duration of each session was 20 min, including warm-up and cool-down, and a rest day in between. The parameters of the instrument used were the following: frequency of 30 Hz, amplitude of 2.5 mm, and five dynamic exercises lasting one minute, followed by 30 s of rest between each exercise. The participants held onto the railing of the platform as a safety measure. The dynamic exercises performed on the vibrating platform were as follows: squat, alternate front knee elevation, hip extension and lowering of the right leg to the ground, hip extension and lowering of the left leg to the ground, and weight shift of the body laterally (Figure 1).

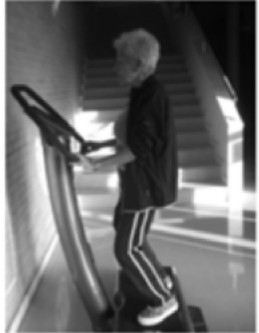

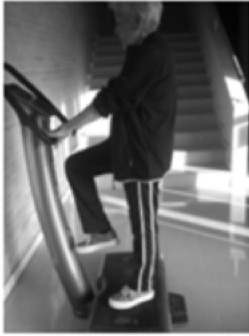
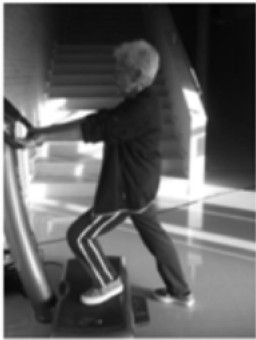

**Figure 1.** Example of dynamic exercises performed on the vibrating platform.

The vibration training was supervised by three physiotherapists; one of them also has a degree in Physical Activity and Sports Sciences. The other two physiotherapists were previously trained in knowledge about WBV, namely, its effects, indications, contraindications, use of vibratory machines, frequencies and amplitudes used in the study, as well as the exercises designed with special incidence to determine the posture they should have in the practical execution of each exercise. Special importance was given to the fact that the ranges of movement of the joints should never exceed the physiological limits of each person and that the exercises that produced pain in the participant would be suppressed until a thorough physiotherapy examination evaluated their origin and aetiology. The parameters of the exercises, like time, intensity or velocity of execution, were performed according to the level of physical fitness of the participants in order to assure good execution of the movement and the correct posture, so as not to cause injuries.

### 2.4. Instruments

2.4.1. Sociodemographic Data Questionnaire

A series of data referring to age, marital status, profession before retirement, number of children, personal history (regarding past or current illnesses and injuries), surgical interventions, and current medication were asked.

2.4.2. Questionnaire to Measure the Quality of Sleep

The Pittsburgh Sleep Quality Index (PSQI) [46] was used to assess the quality of sleep. The version translated into Spanish by Jiménez et al. was used [47]. This questionnaire is made up of 19 items in addition to five questions for roommates. The latter are used as additional information but do not contribute to the total score of the index. The items analyse different determinants of sleep quality grouped into seven components: sleep quality, latency, duration, efficiency and disturbances, the use of sleep medication and daytime dysfunction. Each component is scored from zero to three. From the sum of the seven components, the total score is obtained, which ranges from zero (ease of sleeping) to 21 points (severe difficulty) in all areas. Buysse et al. [46] propose a cut-off point of 5 (score $\geq 5$ defines bad sleepers). The total value of the scale was extracted, as well as Item 7, which refers to drug consumption, and Item 4, which refers to total hours of sleep.

The Spanish version of the PSQI is an appropriate instrument for epidemiological and clinical investigation of sleep disorders. It has good external consistency and an adequate degree of reliability and validity [48]. It also turns out to be an appropriate instrument for the investigation of sleep disorders in geriatric populations [10]. In this study, we used the translation that is faithful to the original since other versions were subjected to some modifications in the adaptation process, which can lead to confusion since even the order of the questions was modified. Although it is a self-administered questionnaire, it was completed by an evaluator who explained each item to the participants. In this study, the values of reliability were $\alpha = 0.79$ (pretest) and $\alpha = 0.91$ (post-test).

2.4.3. Evaluation and Training Material

A calibrated portable stadiometer (Seca 213, Barcelona, Spain) was used for the measurement of stature and weight with bioelectric impedance (Tanita SC 330 S portable, Tanita Corporation, Japan). In the training sessions, DKN XG 3.0 (USA) vibrating machines were used. This type of vibrating platform produces a mechanical stimulus that is characterised by a vertical movement. Its console allows the vibration parameters to be easily selected, having a frequency that oscillates between 20 and 50 Hz. Finally, for the physical exercise sessions, there were two rooms, progressive resistance bands, balls, rings, mats, dumbbells and flotation cylinders. Both rooms had a temperature that ranged between 22° and 24°.

### 2.5. Ethical Considerations

All of the candidates were informed in detail of the characteristics and development of the study, indicating the conditions of participation, with voluntariness and personal disposition for inclusion being an essential requirement, which was indicated by signing an informed consent form for voluntary participation. The deontological standards recognised by the Declaration of Helsinki (64th General Assembly, Fortaleza, Brazil, October 2013) were respected, and the project was approved by the Research Ethics Committee of Jaén (Act 2/2013) as well as by the Committee of Bioethics of the University of Jaén.

### 2.6. Data Analysis

All statistical calculations were performed using the SPSS package (IBM® SPSS® Statistics V22.0), and the statistical difference was established for $p < 0.05$. Homogeneity between the control

and experimental groups was previously verified by Levene's test, assuming that they were equal. Furthermore, the Kolmogorov–Smirnov test showed a normal distribution of the data. Mean, standard deviation, and correlations between variables were analysed using Pearson's correlation, with a value of significance of $p < 0.05$ (*) or $p < 0.001$ (**).

Subsequently, to assess the influence of the treatment on the groups, within the groups and between them, multiple univariate analyses (ANOVAs) were used; first, to show the differences in pretest, and secondly, to show the differences in post-test. After that, Student's *t*-statistic to related samples were used for CG and EG after separating both groups in the database. Finally, we did an analysis consisting of repeated measures on the three variables; the intrasubject factors were "Test" (with two levels: pretest and post-test) and "Group" (with two levels: control and experimental). Likewise, the size of the treatment effect was assessed with Cohen's d [49], considering the values of 0.2–0.5 (small), 0.51–0.8 (moderate), 0.81–1.2 (large), and greater than 1.2 (very large).

## 3. Results

Table 2 shows the summarised behaviour of the variables analysed in the pre- and post-treatment phases for CG and EG and the total number of individuals. The variables "hours of sleep", "Pittsburgh index" and "drug consumption" were considered, which constitute the basic variables of this study in relation to the designed experiment.

**Table 2.** Mean and standard deviation of the variables hours of sleep, Pittsburgh index and drug consumption in the pre- and post-treatment phases and behaviour in pre- and post-treatment between groups.

| | | Pretest | Post-Test | | | Pre–Post Test between Groups | |
|---|---|---|---|---|---|---|---|
| | | Mean ± SD | Mean ± SD | *p*-Value | Size-Effect | F | *p*-Value |
| Hours of sleep | Control | 7.23 ± 1.25 | 6.88 ± 1.03 | 0.083 | −0.27 | 2.307 | 0.135 |
| | Experimental | 6.33 ± 1.12 | 6.94 ± 1.10 | 0.002 ** | 0.53 | | |
| | *p*-value | 0.008 ** | 0.866 | | | | |
| Pittsburgh index | Control | 8.08 ± 4.25 | 10.54 ± 3.76 | 0.001 ** | 0.56 | 6.581 | 0.013 * |
| | Experimental | 8.62 ± 3.99 | 4.77 ± 3.76 | 0.001 ** | −0.94 | | |
| | *p*-value | 0.640 | 0.001 ** | | | | |
| Drug consumption | Control | 1.50 ± 1.33 | 2.08 ± 1.02 | 0.008 ** | 0.49 | 2.380 | 0.129 |
| | Experimental | 1.54 ± 1.36 | 1.07 ± 1.27 | 0.056 | −0.36 | | |
| | *p*-value | 0.919 | 0.003 ** | | | | |

\* $p \leq 0.05$, ** $p \leq 0.01$. SD = standard deviation.

In an initial analysis of the possible relationship between the variable hours of sleep, drug consumption, and the Pittsburgh index, we found that these are related in all cases ($p < 0.001$), finding a significant correlation ($p < 0.001$) in both the pretest and post-test except in sleeping_hours_pre with drugs consumption and Pittsburg_index_post (Table 3).

**Table 3.** Mean, standard deviation and correlations between the variables.

| | | Mean | SD | 2 | 3 | 4 | 5 | 6 |
|---|---|---|---|---|---|---|---|---|
| 1 | Sleeping_Hours_pre | 6.78 | 1.26 | 0.600 ** | −0.237 | −0.058 | −0.676 ** | −0.100 |
| 2 | Sleeping_Hours_pos | 6.91 | 1.05 | | −0.282 * | −0.289 * | −0.560 ** | −0.573 ** |
| 3 | Drugs_consumption_pre | 1.52 | 1.34 | | | 0.560 ** | 0.634 ** | 0.383 ** |
| 4 | Drugs_consumption_post | 1.58 | 1.24 | | | | 0.458 ** | 0.760 ** |
| 5 | Pittsburgh index_pre | 8.35 | 4.09 | | | | | 0.541 ** |
| 6 | Pittsburgh index_post | 7.65 | 4.72 | | | | | |

\* $p \leq 0.05$, ** $p \leq 0.01$. SD = standard deviation.

An initial univariate analysis (ANOVAs) was performed to show the differences in the pretest for both groups. The behaviour of the variables studied before applying the treatment allows us to anticipate that although there are no differences in terms of the Pittsburgh index in CG and EG ($p = 0.640$) or with respect to drug consumption ($p = 0.919$), there was a difference in the behaviour of the hours of sleep between these groups ($p = 0.008$) in favour of CG (more hours of sleep in the CG), although, on average, this difference is not excessive. On the other hand, ANOVAs in the post-test showed differences of $p < 0.001$ on the Pittsburgh index and $p = 0.003$ on drug consumption, both in favour of CG (higher values of "low sleep quality" and "high drugs consumption").

In CG, comparing the pre- and post-treatment phases, we found that there was no difference for the variable hours of sleep ($p = 0.083$, with a small value of Cohen's d), but we did perceive a significant difference between both phases for the Pittsburgh index ($p < 0.001$, with a d value of 0.56). In turn, it is noteworthy that CG has a statistically significant difference ($p = 0.008$) in drug use, showing an increase after the intervention. This difference is explained by an increase in the mean Pittsburgh index in the post-treatment phase compared to the pretreatment phase and also for drug consumption, increasing at both times (mean difference of 2.46 and 0.58, respectively).

In EG, we observed significant differences in both the variable hours of sleep ($p = 0.002$) and the Pittsburgh index ($p < 0.001$). Cohen's d values the effect size between medium (0.53 for the variable hours of sleep and $-0.36$ for drug consumption) and large ($-0.94$ for the Pittsburgh index). The intervals for the difference in means report that these differences are due to an increase in sleep hours, a reduction in drug use, and a reduction in the Pittsburgh index (mean difference 0.61, $-0.47$, 3.85, respectively). On the other hand, a repetitive-measures (MANOVA) was performed in order to show the differences between groups and in the pretest and post-test. The values of repetitive-measured values were significantly different in the Pittsburgh index (f = 6.581; $p = 0.13$) in favour of CG, but not in sleep hours (f = 2.307; $p = 0.135$) or drugs consumption (f = 2.380; $p = 0.129$). Figure 2 and Figure 3 represent the hours of sleep and Pittsburg index, respectively, in CG and EG in the pre- and post-treatment phases.

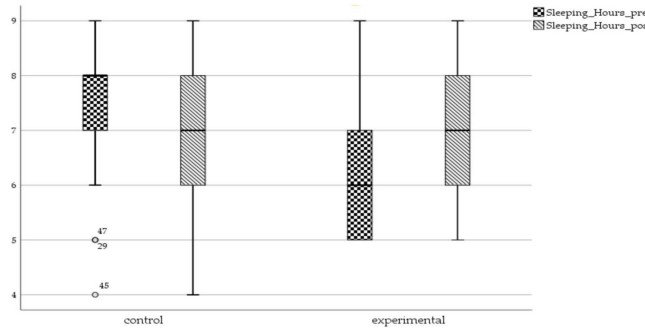

**Figure 2.** Box whisker graphic of sleeping hours in control and experimental groups in pre- and post-treatment phases.

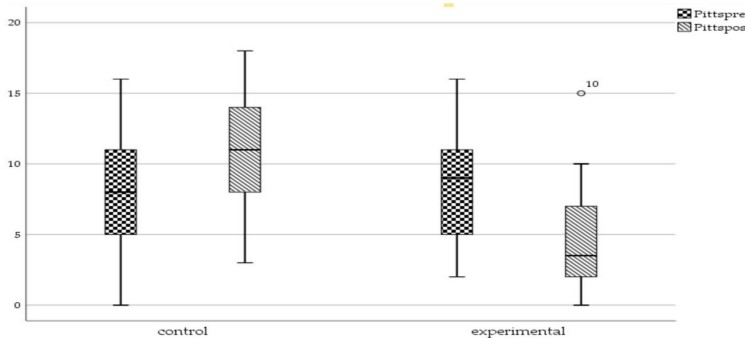

**Figure 3.** Box whisker graphic of the Pittsburgh index in control and experimental groups (pre- and post-test values).

## 4. Discussion

The objective of this study is to evaluate the effects of a regular physical exercise program, complementing it with vibration training, on sleep quality, as well as to analyse the need for drug use to be able to sleep in women over 65 years of age after the intervention was carried out. The main results achieved in this study were an improvement in sleep quality and lower consumption of sleeping drugs in EG.

The optimal results achieved by EG can hardly be compared with other research carried out on this type of exercise. This is because, after reviewing the studies, few have been found that focus on assessing the quality of sleep after WBV except for that of Lin et al. [28], where no improvements were found despite the fact that various authors establish a relationship between physical exercise and sleep quality in the literature [43,50].

First, considering the Pittsburgh index, the results of our study differ from those of Lin et al. [28] who, after an intervention with WBV in people older than 65 years, found improvements in aspects related to sarcopenia, physical capacity and activities of daily living, but not in the quality of sleep. In turn, Marupuru et al. [51] carried out physical training in elderly people, with the same timing and the same instrument used by us, but did not obtain significant results with regard to improving sleep.

Although it has been speculated that vibration can cause possible sleep disturbances, in our research, after each WBV session, participants were asked about their experience with performing the exercises on the platform. All agreed that they felt a pleasant fatigue, nothing bothersome, and never an effort above the bearable threshold. Therefore, vibration does not have to present adverse effects and improvements in sleep quality were obtained. On the other hand, in the review by Lai et al. [27], with more than 30 intervention studies, it was identified that both resistance training and vibration training are beneficial for increasing physical performance in older people. Furthermore, WBV is presented as more recommendable than other programs for the improvement of lung capacity and quality of life in more vulnerable populations [52]. Indeed, the results of Wadsworth and Lark [53] indicate that 16 weeks of low-WBV exercise provides an adequate and easily accessible stimulus for frail elderly people to achieve improved levels of physical function.

On the other hand, CG only performed physical exercise, which led to reduced values in our study with respect to the index of sleep quality and the hours of sleep. These results are in contrast to those of other authors since, as some experts point out, physical exercise performed regularly can lead to an improvement in insomnia symptoms, related to the initiation and maintenance of night-time sleep, in people over 65 years [54,55]. Although only two days of weekly physical exercise were performed in this study, various authors have indicated that the frequency of exercise is essential to improve the quality of sleep [19,56], as no improvements in the quality of sleep have been found with a greater number of sessions in older people [19]. Moreover, in our intervention, the cool-down period may not have been enough to produce a sense of relationship in the participants. It would have been interesting to include breathing exercises, some yoga postures or Tai Chi movements.

We contrast these results with those of researchers who identified that the change from sedentary lifestyle to low-intensity exercise could have long-term benefits on the quality of sleep of the elderly [15]. Although Choi and Sohng [57] reported that subjects reported increased muscle strength and shoulder flexibility and reduced symptoms of depression after a 12-week sitting yoga program, it had no significant effect on sleep quality. According to the study by Kakinami et al. [58], with young adults, no relationship was found between sedentary time and sleep quality.

In contrast to the above, the results of our analysis provide support to the hypothetical components of sleep quality and, in a singular way, separate the quantitative from the qualitative aspects. Furthermore, the relevance in distinguishing these two aspects was confirmed by the absence of significant differences between the groups regarding the duration of sleep after the intervention, although it increased in EG without changing in CG. This finding underscores how inaccurate and incomplete an assessment of sleep based solely on quantitative aspects, such as the number of hours of

sleep per night, can be [47]. In our study, the Pittsburgh index variable showed significant differences, with a lower value in EG compared to CG.

As we already pointed out, vibration causes a neurogenic effect of vasodilation, increasing blood flow to the muscles and reducing their acidification [31,32]. It also influences the production of hormones [33] and reduces the consumption of drugs in some age-related diseases [34]. The vibration also produces a relaxing effect, which the participants reported after it was performed. Among its effects at the neuromuscular level, it produces an increase in the flexibility of the muscles, which also increases the feeling of relaxation.

It is noteworthy that the activity carried out by the participants of both groups in the present study focused on a component of strength-resistance of moderate–high intensity, which differs from the studies with older people who observed improvements in the quality of sleep through low-intensity [14,54] or moderate-intensity physical exercise [19]. In contrast, Aguilar-Parra et al. [59] reported that a physical exercise program, similar to the one in the present study, improved sleep quality and reduced insomnia. The same results were found by Jaechul [56], especially in older women.

As a second objective, we must also point out that EG, in addition to improving sleep quality, was the only one of the two groups that reduced the consumption of sleeping drugs. The reasons for which CG increased its consumption are not entirely clear since certain studies have shown that physical activity can reduce the need for medication to be able to sleep [44,60–63]. In contrast, in our study, only the combination of both interventions (physical exercise + WBV) produced a decrease in the said consumption. Previously, we found a decrease in pain and an improvement in the quality of life with this combined protocol of physical exercise and WBV [64] that can explain these results.

Thus, the combined procedure of these two intervention strategies, with exercise and vibration, seems to produce an improvement in the quality of sleep of older women, both from quantitative (hours of sleep) and qualitative points of view (Pittsburgh index). However, the physical exercise program alone did not show benefits in the quality of sleep; even the perception of satisfaction with sleep decreased. There is a possible clinically relevant beneficial effect of WBV in addition to physical exercise. This modality of intervention can favour the quality of life of the elderly by improving all of the organic processes that take place during sleep and improving activities in the waking state.

As limitations, we can point out that a long-term follow-up has not been carried out after the intervention, but a high level of therapeutic adherence can be verified (with only dropouts in the control group, which we understand is due to the exhaustive follow-up in both groups). Consistent with Mikhael et al. [65], there is a greater magnitude of the effect of the intervention with complete control and follow-up, due to the possibility of the therapist adjusting the individual's progress within the program. In addition, control of the diet and the amount of liquid at certain hours of the night would have been interesting since they could have a potential additional effect on the treatment used in this group of older people.

Several studies have proven that sleep quality is related to circadian rhythm disorders, specific sleep disorders, and use of medication or clinical diseases, since, according to certain studies, approximately 80% of the elderly who are 70 years and older have at least one of these seven clinical diseases: high blood pressure, cardiovascular diseases, arthritis, diabetes mellitus, cancer, acute myocardial infarction, or respiratory diseases. In this study, we include people with chronic diseases to improve their health, functional capacity and quality of sleep, as long as they are not in the acute phase of their disease. In the exclusion criteria, it was determined that the medical authorisation to participate should inform of any cardiorespiratory pathology in the acute or unstable phase and, therefore, the participant would have physical exercise contraindicated.

Women report poorer sleep when they perceive adverse psychosocial factors. The adverse aspects of emotional support and social networks contribute significantly to the pattern of bad sleep [66]. Loneliness and personality [67], psychological distress (depression and anxiety) [68], and environmental factors [69] are important in the quality of sleep. We could not control for these psychosocial factors that could have influenced the results.

Therefore, we can state the clinical relevance of the findings that we report, which will serve as a contribution to future studies that will provide more evidence for our research, where the effects of WBV on sleep quality in older adults are analysed. This is because adequate rest has a positive impact on health, quality of life and functional independence, even preventing the development of chronic diseases. The effectiveness of WBV lies in its ease of use, the short time required for use, low aerobic demands and mild effects on blood pressure [70].

Future studies could consider the possibility of carrying out similar trials, experimenting with different intervention modalities in terms of frequency and duration of treatment, as well as intervening in other factors (such as diet) that may influence the results. Additionally, it would be important to have a control group that did not exercise, to know if only WBV can achieve the effect on sleep without prior training.

## 5. Conclusions

Twelve weeks of vibration training, combined with physical exercise, could be an adequate intervention strategy in older women to improve their quality of sleep. Thus, WBV is presented as a form of physical exercise suitable for older people since, among others, the time invested is reduced, it can be performed at home (thereby achieving maximum adaptability to situations such as confinement), it allows the dose to be adapted for each person, and it can also guide the workload. Finally, this type of intervention can help to reduce the consumption of drugs needed to sleep by improving the conciliation of sleep in a natural way.

**Author Contributions:** Conceptualisation, M.A.-C.; data curation, M.V.P.-M. and D.M.-S.; formal analysis, E.L.-A., J.A.P.-M., and D.M.-S.; investigation, M.V.P.-M., E.L.-A., and M.A.-C.; methodology, M.V.P.-M., E.L.-A., V.S.-H., and J.A.P.-M.; supervision, V.S.-H. and J.A.P.-M.; writing—original draft, M.V.P.-M., E.L.-A., V.S.-H., and J.A.P.-M.; writing—review and editing, M.V.P.-M. and D.M.-S. All authors have read and agreed to the published version of the manuscript.

**Funding:** This research received no external funding.

**Conflicts of Interest:** The authors declare no conflict of interest.

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
