# Peer review of "Sleep Quality in Older Women: Effects of a Vibration Training Program"

_applsci, doi:10.3390/app10238391_

Round 1
Reviewer 1 Report
This is a well written paper. However, more information with regard to how "controlled" the CG group was is needed.
a) need more reporting and discussion on diet and other factors that influence sleep and sleep behavior
b) need more information and discussion about the actual efficacy of the exercise with and without vibration on other physical and cognitive changes. It is difficult to decipher the potential mechanisms of the vibration training in addition to the other physical training without knowing exactly how much and how effective the exercise program was on muscular strength and endurance as well as if any CV changes were noted following the aerobic exercise.
c) the total time period, number of sessions and/or the volume of the exercise and exercise plus vibration training performed should be reported.
d) discuss the potential mechanisms for why additive vibration would be superior to regular exercise for reducing drug use and or improving sleep.
e) discuss whether people were participating in other "outside" activities, or had different amounts of physical or emotional stress.
f) discuss whether the vibration training was improved other issues that may have been impacting sleep -- such as pain.
Overall this paper would benefit from a greater and more in-depth discussion of how various factors that could effect sleep in this population were handled and/or controlled.
There are only minor English awkward language style issues: example line 108, line 113 and line 118. These references to numbers are not in the same category.
Author Response
Thanks you for take into account this manuscript. We are tried take into account all your suggestions.
This is a well written paper. However, more information with regard to how "controlled" the CG group was is needed.
This information has been included in the document (Lines 170-174).
Need more reporting and discussion on diet and other factors that influence sleep and sleep behavior
We have selected some factors that they could influence in sleep behavior and the articles have been included (Lines 45-47)
Need more information and discussion about the actual efficacy of the exercise with and without vibration on other physical and cognitive changes. It is difficult to decipher the potential mechanisms of the vibration training in addition to the other physical training without knowing exactly how much and how effective the exercise program was on muscular strength and endurance as well as if any CV changes were noted following the aerobic exercise.
We have included more information about this tips (effect of exercise in sleep quality or physical capacity Lines 66,67). On the other hand, we have not found any study that compare the efficacy of the exercise comparing with and without vibration
c) the total time period, number of sessions and/or the volume of the exercise and exercise plus vibration training performed should be reported.
Done (Lines 154-158)
d) discuss the potential mechanisms for why additive vibration would be superior to regular exercise for reducing drug use and or improving sleep.
Done (Lines 394-397)
e) discuss whether people were participating in other "outside" activities, or had different amounts of physical or emotional stress.
It has been included (Lines 170-174)
f) discuss whether the vibration training was improved other issues that may have been impacting sleep -- such as pain.
It has been included with a reference (Lines 394-397)
Overall this paper would benefit from a greater and more in-depth discussion of how various factors that could effect sleep in this population were handled and/or controlled.
This factors have been included (Lines 414-427). The control of diet is included as limitation (Line 411-413)
There are only minor English awkward language style issues: example line 108, line 113 and line 118. These references to numbers are not in the same category.
We have changed this references, and we have checked with a English company all the manuscript (We attached the certificated document)

Reviewer 2 Report
View file

Author Response
It would be important to have a control group that did not exercise, to know if only vibratory training can achieve the effect on sleep without prior training. It would also be interesting to talk about the psychosocial factors that influence sleep.
We have included as futures studies (Line 436-438). Psychosocial factors with references have been included (Lines 414-427)
On page 109 it should not be ministery is Consejería.
Done (Line 144)
In the methods I think it would be more interesting to put the general training program before the vibratory one.
Done
I think we should provide more results. For example, put a table with the general characteristics of the groups: age, weight, height BMI. I also see that a table is missing that shows the most significant data from the socio-democratic study.
This characteristics have been included with more detail in table 1 (Line 139)
I think it needs a revision of the English.
We have checked the manuscript with an English company (We attached the certificated document)
Thanks you for your appreciations.

Reviewer 3 Report
Sleep quality in older women: Effects of a vibration training program
General
Thank you for giving the opportunity to review the manuscript entitled “Sleep quality in older women: Effects of a vibration training program”. Authors established an interesting research question regarding the effect of an additional WBV training program in older women. Although the research question is interesting and the findings are promising there are some major concerns regarding the experimental procedure of the current study. Thus, I would like to give the opportunity to authors to revised the manuscript as it is always nice to improve the quality of life of older people.
General Comments, Main Concerns
- How did authors suppurate the participants into the CG and EG? It is unclear what the main variable for this separation was.
- Was the WBV performed in different days than the Physical exercise program? Is there a case that the time of WBV may affect the results?
- Why authors used physiotherapist to supervise the WBV exercise program? Isn’t this the job for sport science and physical education specialists?
- Why authors performed T-Test analysis when the study has two groups with pre and post measurements? This really limits the results.
- How authors explain the increase in Pittsburgh index and in Drug consumption for the CG?
Abstract
Abstract is well written and provides a good overall of the study.
Line 17-19: I suggest rephrasing the first lines in order to match the purpose of the study.
Line 24: Here Authors present that both groups performed two training sessions per week but the EG performed 4 training sessions per week including WBV training.
Line 33, Key words: Please, use different key-words other than the title of the manuscript.
Introduction
Introduction is well written presenting the effect of sleep in general, sleep difficulties, the role of physical exercise and drugs. However, the effect of WBV on elderly or in the quality of sleep is not well analysed. Authors present many studies in the discussion section but failed to present these data in the introduction. Moreover, it is unclear why it is important to examine the WBV training stimulus in elderly and how elderly will response to this type of exercise.
In addition, I suggest to Authors to add a research hypothesis in the end of the introduction.
Materials and Methods
Line 87: Question: Why is the reference 24 here?
Line 94: Ages ranged from 65-85. This is a great range of age between participants. How authors ensured the rate of adaptations between the participants? Some exercise might be difficult for 75 and above and easier for 75 and below.
Line 96: Did the Authors examine the participants for heart problems? Although the training program is not composed by isometric exercises, this should be included inside the manuscript.
Line 99: How authors suppurate the participants into CG and EG? This is a serious experimental manipulation for safe results.
Line 105: Please provide a percentage of training sessions that lost during the training period.
Line 124: WBV training is not clear. Authors are kindly suggested to add training volume, repetitions per minute, intensity, velocity of movement and how authors control the progressiveness of the training program during the 12 weeks period.
Line 134: Is 20 minutes training duration enough time for a training session?
Line 137-140: Looking again the exercises in the WBV training program is clear that the lower body was involved. I cannot see any exercise for the upper body, which makes sense since authors had older participants. However, why it is Whole Body Vibration?
Line 141: I am confused here. For the physical exercise program there was a 10 years’ experience Physical Activity and Sports Supervisor while the WBV training was supervised by physiotherapists. This is confusing and it makes me really thinking about the training method. With what certification did the physiotherapists supervised the WBV training?
Line 148: Please provide some details about the temperature of the exercise room and the exact time of the day.
Lines 151-153: I suggest to authors to add a table with the variables of the training program.
Line 159: How did authors collect this data?
Line 164: Although the Pittsburgh Sleep Quality Index is a valid measurement of sleep quality, is there a reliability value to present?
Line 178: Provide a test re-test value (ICC or CV%) for seca 213.
Line 179: Provide the country of origin.
Lines 182-183: That is why I recommended adding a table with the exact training variables. Please include repetitions or time of exercise, intensity, etc.
Line 185: Super.
Line 194: Why authors used the T-Test analysis? How many T-Test did they performed? Instead, why not using the 2 way ANOVA with repeated measures? This really limits your results. In addition, how authors can conclude that there was a difference between groups following the treatment? I strongly suggest to authors to change the statistical analysis of the study.
Similarly, authors present data from correlation. However, correlation is not presented inside the statistics section.
Question: How authors measured drugs consumption? Is missing in the methods.
Results
Table 1: This is tricky here. CG has a T1 value for hours of sleep 7.23 ± 1.25h, while EG has a T1 value of 6.33±1.22h. Is there a difference between groups during the initial measurements? ANOVA will be able to give that answer.
Line 213-214: Please add correlation analysis inside the statistics. Furthermore, please provide the r value for the significant correlations and not only the p values.
Lines 216-222: How authors calculate these values? Please clarify.
Line 236-245: Why it is important to present the results for both groups? There was a different training intervention to EG and a different training intervention to the CG.
Figures 2-3: Please add legends on the side of the figures.
Discussion
Line 258: I suggest to authors to present the main findings of the study. What was the main outcome of this manuscript?
Line 279: How authors explain the reduced values in sleep quality and hours of sleep in the CG? What happen to the participants that might explain this result? As authors mention inside the discussion these results are in contrast with other studies. Is this because authors used females? Moreover, comparing with the EG the initial values at least in hours of sleep seems to be greater than CG. A good explanation here would be more interesting for the reader.
Lines 307-310: Here is another un-answered question. Why CG increased sleeping drugs although the participation in physical exercise program? I am sure that after a 12 weeks training project authors collected a lot of data. For both question authors may search inside their data or for blood indicators in order to explain these results.
Conclusions
Nice take home message.
Line 340: WBV can be performed in home?
Author Response
General
Thank you for giving the opportunity to review the manuscript entitled “Sleep quality in older women: Effects of a vibration training program”. Authors established an interesting research question regarding the effect of an additional WBV training program in older women. Although the research question is interesting and the findings are promising there are some major concerns regarding the experimental procedure of the current study. Thus, I would like to give the opportunity to authors to revised the manuscript as it is always nice to improve the quality of life of older people.
Thanks you for your words and your exhaustive review to improve this manuscript.
General Comments, Main Concerns
How did authors suppurate the participants into the CG and EG? It is unclear what the main variable for this separation was.
It has been clarified (Lines 130-133)
Was the WBV performed in different days than the Physical exercise program? Is there a case that the time of WBV may affect the results?
It has been included (Lines 154-158). We have included as possible future line that the time of WBV may affect the results (there are not references using different time of WBV and physical activity on sleeping quality. Lines 434-438)
Why authors used physiotherapist to supervise the WBV exercise program? Isn’t this the job for sport science and physical education specialists?
We have explain it in the manuscript (Lines 193-201). A physiotherapist is graduate in Physical Activity and Sport Sciences.
Why authors performed T-Test analysis when the study has two groups with pre and post measurements? This really limits the results.
We have made the statistical analysis using a more powerful test. This have been described with more detail (line 260-267) and the results have been included in all Results section (table 2 and 3, Lines 278-280 and 287-288). On the other hand, we used a Multivariate analysis (MANOVA) but the results of multivariate effect had not significant statistical effect, since the purpose of the study was to show the differences in only 3 variables, we decided include only the ANOVA results.
How authors explain the increase in Pittsburgh index and in Drug consumption for the CG?
It is an interested question. Maybe it could be because the vibrations provide a greater relaxing effect, which the participants reported after they were performed. Its effects at the neuromuscular level produce greater flexibility of the muscles that also increases the feeling of relaxation (Lines 378-383). Additionally, there are other factors that influence the quality of sleep that we could not control and that will be reflected in our limitations (nutrition, stress control, social and psychological factors Lines 414-427).
Abstract
Abstract is well written and provides a good overall of the study.
Line 17-19: I suggest rephrasing the first lines in order to match the purpose of the study.
Done (Lines 19,20)
Line 24: Here Authors present that both groups performed two training sessions per week but the EG performed 4 training sessions per week including WBV training.
It has been clarified (Lines 25-27)
Line 33, Key words: Please, use different key-words other than the title of the manuscript.
Done (Lines 36,37)
Introduction
Introduction is well written presenting the effect of sleep in general, sleep difficulties, the role of physical exercise and drugs. However, the effect of WBV on elderly or in the quality of sleep is not well analysed. Authors present many studies in the discussion section but failed to present these data in the introduction. Moreover, it is unclear why it is important to examine the WBV training stimulus in elderly and how elderly will response to this type of exercise.
Done (Lines 81-84 and 88-95)
In addition, I suggest to Authors to add a research hypothesis in the end of the introduction.
It has been included (Lines 104-109)
Materials and Methods
Line 87: Question: Why is the reference 24 here?
It was a mistake, we have deleted this reference
Line 94: Ages ranged from 65-85. This is a great range of age between participants. How authors ensured the rate of adaptations between the participants? Some exercise might be difficult for 75 and above and easier for 75 and below.
This is the age range of the elderly who go to the day center for physical exercise. Surprisingly 85-year-old women achieved better test results. Here we discover that biological age does not have to coincide with physiological age. It all depends on the functional capacity of each person.
Line 96: Did the Authors examine the participants for heart problems? Although the training program is not composed by isometric exercises, this should be included inside the manuscript.
Thanks for your consideration. In the exclusion criteria, we will specify that the participants provided us with the medical report of their personal history and current pathology. Women with acute or decompensated heart and respiratory diseases were not included in our study (Lines 124-126)
Line 99: How authors suppurate the participants into CG and EG? This is a serious experimental manipulation for safe results.
Done (Lines 130-133)
Line 105: Please provide a percentage of training sessions that lost during the training period.
Done (Lines 134-136)
Line 124: WBV training is not clear. Authors are kindly suggested to add training volume, repetitions per minute, intensity, velocity of movement and how authors control the
Movement speed was not measured. Each participant performed the exercises according to their ability (we were more interested in the good execution of the movement and the correct posture so as not to cause injuries).
Line 134: Is 20 minutes training duration enough time for a training session?
We considered that it could be adequate following the references from Ritweger et al (Lines 432-433)
Line 137-140: Looking again the exercises in the WBV training program is clear that the lower body was involved. I cannot see any exercise for the upper body, which makes sense since authors had older participants. However, why it is Whole Body Vibration?
Indeed, the exercises were programmed for the lower limbs, but the participants had to hold onto the platform railing to perform them, which involved an additional grip force. WBV produces neuronal activation at the peripheral level and also at the cortical level.
Line 141: I am confused here. For the physical exercise program there was a 10 years’ experience Physical Activity and Sports Supervisor while the WBV training was supervised by physiotherapists. This is confusing and it makes me really thinking about the training method. With what certification did the physiotherapists supervised the WBV training?
One of the physiotherapists also has a degree in Physical Activity and Sports Sciences. He instructed the others in the program developed by another graduate in Physical Activity and Sports. The other physiotherapists observed the correct posture of each participant and if they presented any type of pain or discomfort during the execution of the exercises (Lines 193-201).
Line 148: Please provide some details about the temperature of the exercise room and the exact time of the day.
Done (Lines 158 and 243-244)
Lines 151-153: I suggest to authors to add a table with the variables of the training program.
We have not included this table because the variables like time, intensity or frequency of exercises it was depended of the participant (we tried to control the variables individualizing the intervention as much as possible according to their physical condition)
Line 159: How did authors collect this data?
The physical exercise program was attended by the main researcher to see its development and ask the participants about their level of satisfaction and possible discomfort.
The person in charge of carrying out the program took the roll at the beginning of the sessions and then gave the researcher the list of participants in each session.
It have been included (Lines 170-174)
Line 164: Although the Pittsburgh Sleep Quality Index is a valid measurement of sleep quality, is there a reliability value to present?
This text had been included (lines 227-234). To present study we have included the reliability value (line 234)
Line 178: Provide a test re-test value (ICC or CV%) for seca 213.
We have not the value of icc-cv%. We follow the instruction in order to caliber the stadiometer Seca 213
Line 179: Provide the country of origin.
It have been included (Line 237).
Lines 182-183: That is why I recommended adding a table with the exact training variables. Please include repetitions or time of exercise, intensity, etc.
The explication about this point have been included in lines 193-203
Line 185: Super.
Line 194: Why authors used the T-Test analysis? How many T-Test did they performed? Instead, why not using the 2 way ANOVA with repeated measures? This really limits your results. In addition, how authors can conclude that there was a difference between groups following the treatment? I strongly suggest to authors to change the statistical analysis of the study.
We have made the statistical analysis using a more powerful test. This have been described with more details in lines (260-268) and the results have been included in all Results section (table 2 and 3, 278-289)
Similarly, authors present data from correlation. However, correlation is not presented inside the statistics section.
This have been included (table 3, line 287-288).
Question: How authors measured drugs consumption? Is missing in the methods.
The Pittsburgh Sleep Quality Index (PSQI) was used to assess the quality of sleep. The total value of the scale was extracted, as well as item 7 that refers to drug consumption and item 4 that refers to total hours of sleep (Lines 225-226)
Results
Table 1: This is tricky here. CG has a T1 value for hours of sleep 7.23 ± 1.25h, while EG has a T1 value of 6.33±1.22h. Is there a difference between groups during the initial measurements? ANOVA will be able to give that answer.
It is included below the table 3 (lines 290-297). It has been changed to improved the comprehension
Line 213-214: Please add correlation analysis inside the statistics. Furthermore, please provide the r value for the significant correlations and not only the p values.
It have been included below table 3 (line 282-290)
Lines 216-222: How authors calculate these values? Please clarify.
It was the ANOVA test values that was performed to show the differences between both groups in the pre-test. It was the multiple univariate analysis (ANOVAs) test that was performed to show the differences between both groups in the pre-test. It has been included on the table 3
Line 236-245: Why it is important to present the results for both groups? There was a different training intervention to EG and a different training intervention to the CG.
We have deleted the complete group analysis.
Figures 2-3: Please add legends on the side of the figures.
Legends are included in both figures (in the upper right corner)
Discussion
Line 258: I suggest to authors to present the main findings of the study. What was the main outcome of this manuscript?
Done (Lines 329-331)
Line 279: How authors explain the reduced values in sleep quality and hours of sleep in the CG? What happen to the participants that might explain this result? As authors mention inside the discussion these results are in contrast with other studies. Is this because authors used females? Moreover, comparing with the EG the initial values at least in hours of sleep seems to be greater than CG. A good explanation here would be more interesting for the reader.
We have specified it in the lines 361-363, 378-383 and 414-427.
Lines 307-310: Here is another un-answered question. Why CG increased sleeping drugs although the participation in physical exercise program? I am sure that after a 12 weeks training project authors collected a lot of data. For both question authors may search inside their data or for blood indicators in order to explain these results.
We have dates but regarding systolic and diastolic pressure and arterial pressure it was not modified in the control group. On the other hand, the bilateral ankle swelling increasing after the intervention in the control group but we do not know with certainty what this increase in drug use could have been due to.
Conclusions
Nice take home message.
Line 340: WBV can be performed in home?
Yes, and it could be really useful because you don’t need to go to another place for instance. Although we recommend that the program be supervised by a graduate in Physical Activity and Sports.
Thanks you for your comments. we hope we have attended to all your considerations properly

Round 2
Reviewer 3 Report
No comments
Author Response
Thanks you, we consider we have improve this manuscript thanks to your appreciations.